# Basic Mechanical Properties of Self-Compacting Concrete Prepared with Aeolian Sand and Recycled Coarse Aggregate

Shiqi Zheng [1], Qing Liu [2,*], Fengxia Han [2], Shan Liu [2], Tong Han [2] and Hao Yan [1]

1    College of Architectural and Civil Engineering, Xinjiang University, Urumqi 830047, China; 107552101534@stu.xju.edu.cn (S.Z.)

2    Key Laboratory of Building Structure and Seismic Resistance of Xinjiang, Urumqi 830017, China

*    Correspondence: liuqing2666@xju.edu.cn

**Abstract:** To protect the environment and reduce the consumption of natural resources, this study investigated the performance of self-compacting concrete (SCC) utilizing aeolian sand (AS) as a replacement for fine aggregate and recycled coarse aggregate (RCA) as a replacement for natural coarse aggregate. Twelve mixing ratios were prepared, with AS replacement ratios at 0, 20%, 40%, and 60% and RCA replacement ratios at 0, 25%, and 50%. The evaluation primarily focused on workability, uniaxial compressive strength, split tensile strength, stress–strain curve, modulus of elasticity, and axial compressive strength. The results indicated that both AS and RCA can enhance the performance of SCC at appropriate replacement ratios, and a 20% AS and 50% RCA replacement ratio significantly improved the performance of SCC. In addition, the effects of AS and RCA replacements on SCC were evaluated by several performance indexes, which provides a basis for subsequent experimental studies and demonstrates the feasibility of incorporating AS and RCA into SCC.

**Keywords:** self-compacting concrete; aeolian sand; recycled coarse aggregate; mechanical properties; workability





## 1. Introduction

Concrete is one of the most widely used engineering materials in the construction industry due to its mechanical properties, durability, and cost-effectiveness [1,2]. A large amount of natural resources have been consumed as fine and coarse aggregates. Consequently, many researchers have explored alternative materials for sand and coarse aggregates. Self-compacting concrete (SCC), developed in 1988, should meet three basic requirements in its fresh state: filling capacity, passage capacity, and resistance to segregation [3–6]. These characteristics are achieved through the use of aggregates and chemical admixtures.

The disposal of damaged or end-of-life concrete contributes significantly to construction and demolition (C&D) waste [7,8]. Studies have demonstrated that the material properties of recycled coarse aggregate (RCA), including workability [9–11], mechanical properties [12–14], and durability [15–17], are generally inferior to those of natural coarse aggregate (NCA) due to the presence of a considerable number of microcracks in RCA, which occur during both the production and recycling processes [18]. SCC with recycled aggregates has been extensively and deeply studied. Despite some drawbacks, incorporating recycled aggregates into SCC is feasible. Many researchers have confirmed that recycled concrete meets the requirements of workability and mechanical property when the replacement ratio of coarse aggregate is less than 50% [19–23].

Desertification, a major global ecological issue, affects human survival and development. Deserts account for approximately one-third of the Earth's land area and about 16% of China's land area [24]. Aeolian sand (AS), primarily sourced from deserts and desertification areas, provides the main source of sand for sandstorm activities, which

significantly impacts the ecological environment. The study of incorporating AS in concrete has practical significance for ecological protection and sustainable development [25,26]. Although AS has traditionally been unsuitable for concrete due to its gradation and other reasons, recent studies have confirmed that AS can be incorporated into concrete as a fine aggregate based on its chemical composition and engineering properties [27–30]. However, as the composition of grains and minerals varies between different deserts and even between different sections within the same desert, the regional characteristics of AS vary significantly [27,31–34]. Thus, it is necessary to conduct research on concrete with AS in the Xinjiang area.

Despite extensive studies on the addition of RCA or AS into concrete, there are few studies on their combined use under the unique natural conditions in Xinjiang. Therefore, it is essential to conduct research on self-compacting concrete with AS and RCA in Xinjiang, China. It is not clear how they jointly affect the physico-mechanical properties of concrete. To address this issue, this study designed different mixing ratios of AS and RCA and conducted various workability and physico-mechanical tests to determine the optimal mixing ratios. Microstructural characteristics of concrete under different mixing ratios were analyzed using scanning electron microscopy (SEM) and mercury intrusion porosimetry (MIP) tests to explore their influence on the macroscopic physico-mechanical properties.

The primary objective of this research was to examine the impact of AS and RCA on the fresh and basic mechanical properties of SCC at different replacement ratios. In addition, the effects of AS and RCA replacement ratios on microstructural changes in aeolian sand recycled coarse aggregate self-compacting concrete (ARSCC) were investigated. The results of the study showed that both AS and RCA improved the workability and basic mechanical properties of ARSCC at appropriate replacement ratios.

## 2. Materials and Methods

### 2.1. Material Properties

2.1.1. Binders, Water, and Admixtures

In the production of concrete mixtures, PO42.5 Portland cement, which conforms to the standard GB 175-2007 [35], and fly ash, which conforms to the standard GB/T 1596-2017 [36], were used for all concrete mixtures as a binder. The strength grade of cubic concrete specimens prepared in this study was C30. The characteristic properties of Portland cement are summarized in Table 1, the chemical composition of fly ash is presented in Table 2, and the performance parameters are shown in Table 3.

**Table 1.** Basic properties of cement.

| Cement Stability | Fineness (Sieve Size: 80 μm)/% | Specific Surface (m²/kg) | Setting Time (min) | | Compressive Strength (MPa) | |
|---|---|---|---|---|---|---|
| | | | Initial | Final | 3 d | 28 d |
| Qualified | 0.6 | 360 | 85 | 210 | 27.8 | 48 |

**Table 2.** Chemical composition of fly ash (%).

| Ingredient | $SiO_2$ | $Al_2O_3$ | $Fe_2O_3$ | CaO | $Na_2O$ | MgO | $K_2O$ | $SO_3$ |
|---|---|---|---|---|---|---|---|---|
| Content | 50.79 | 15.92 | 11.10 | 10.29 | 3.52 | 2.60 | 2.22 | 1.43 |

**Table 3.** Performance parameters of fly ash.

| Fineness (%) | Moisture Content (%) | Water Demand Ratio (%) | Ignition Loss Rate (%) |
|---|---|---|---|
| 22.2 | 0.1 | 91 | 3.1 |

The water used in this study comes from Urumqi, and it is considered potable and is generally used for manufacturing concrete.

The water reducer with a water reduction rate of about 25–30% is an important additive for ensuring workability in SCC.

### 2.1.2. Fine Aggregate

The aeolian sand (AS) employed in this study was sourced from the Gurbantunggut Desert, and the natural sand (NS) was taken from Urumqi West Construction Ltd. (Urumqi, China). The chemical composition of AS and NS is mainly $SiO_2$ and $Al_2O_3$ without harmful composition to concrete, as shown in Table 4. The basic physical parameters of the fine aggregates used in this study are shown in Table 5. The particle size distribution of the fine aggregates obtained from the sieve test is shown in Figure 1. The gradation of NS meets the requirements of the regulation limit in the Chinese specification, while the gradation of AS is lower than that of the regulation limit.

**Table 4.** Chemical composition of fine aggregate.

| Samples | $SiO_2$ | $Al_2O_3$ | CaO | $Fe_2O_3$ | $Na_2O$ | $K_2O$ | MgO | $TiO_2$ | $P_2O_5$ | $SO_3$ | Other |
|---------|---------|-----------|------|-----------|---------|--------|------|---------|----------|--------|-------|
| AS | 75.09 | 11.16 | 3.66 | 3.13 | 2.68 | 2.34 | 0.96 | 0.65 | 0.16 | 0.03 | 0.14 |
| NS | 90.76 | 4.59 | 0.11 | 0.73 | 0.39 | 2.2 | 0.18 | 0.14 | 0.07 | 0.07 | 0.76 |

**Table 5.** Basic physical properties of fine aggregate.

| Sand | Source | Technical Indicators | | |
|------|--------|----------------------|---|---|
| | | Fineness Modulus | Soil Clay Content (%) | Apparent Density (kg/m$^3$) |
| AS | Gurbantunggut Desert | 0.12 | 2.60 | 2542.40 |
| NS | Urumqi sand field | 2.93 | 0.80 | 2487.50 |

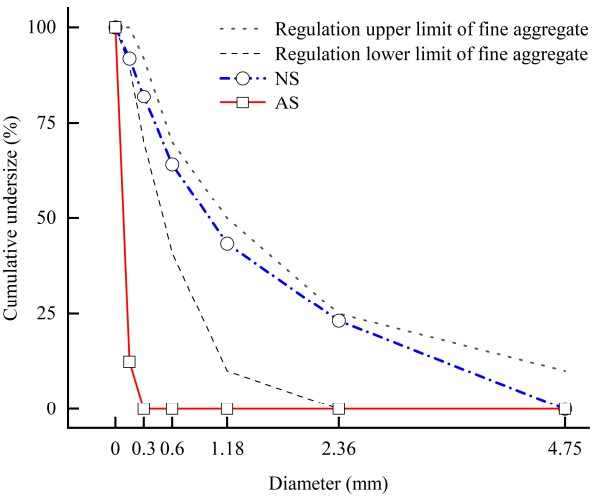

**Figure 1.** Particle size distributions of fine aggregate.

### 2.1.3. Coarse Aggregate

Recycled coarse aggregate (RCA) was obtained from concrete roads that were demolished in Urumqi, while natural coarse aggregate (NCA) was taken from Urumqi West Construction Ltd. This study defines RCA as a fraction retained between 2.36 and 19 mm sieve. The basic physical parameters of coarse aggregate used in this study are shown in Table 6. The particle size distributions of coarse aggregate derived by sieve analysis are depicted in Figure 2.

**Table 6.** Properties of coarse aggregate.

| Aggregate Category | Water Absorption (%) | Moisture Content (%) | Packing Density (kg/m³) | Apparent Density (kg/m³) | Elongated Particle (%) | Crush Index (%) |
|---|---|---|---|---|---|---|
| NCA | 2.36 | 0.81 | 1204.27 | 2485.25 | 1.58 | 10.44 |
| RCA | 3.60 | 0.40 | 1353.00 | 2687.22 | 3.15 | 8.76 |

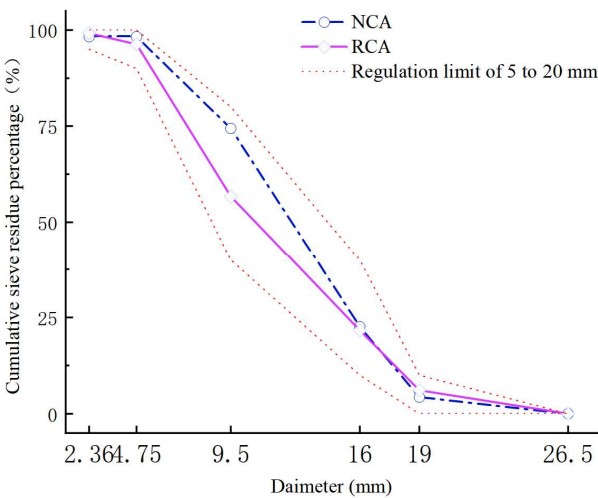

**Figure 2.** Particle size distributions of coarse aggregate.

### 2.2. Mix Proportions of ARSCC

In this study, the design of mixing ratios was used for the basic mechanical properties study, based on the previous experimental studies [37–39]. The previous research results and conclusions about aeolian sand and recycled aggregates indicate that the substitution rate for self-compacting concrete mixed with AS should not exceed 60% to obtain better mechanical and durability performances, whereas that for RCA is recommended to not exceed 50% to ensure its strength and practicality [20,40]. We designed an experimental mix with a replacement ratio of aeolian sand and recycled coarse aggregate, as shown in Table 7.

**Table 7.** Mix design (kg/m³).

| Mix Code | Fly Ash | Cement | RCA | NCA | AS | NS | Water | W/B | Added Water | WR |
|---|---|---|---|---|---|---|---|---|---|---|
| A0-R0 | 259.37 | 257.71 | 0.00 | 848.00 | 0.00 | 706.86 | 169.95 | 0.33 | 0.00 | 3.33 |
| A0-R25 | 259.37 | 257.71 | 199.94 | 636.00 | 0.00 | 706.86 | 169.95 | 0.33 | 13.89 | 3.33 |
| A0-R50 | 259.37 | 257.71 | 399.89 | 424.00 | 0.00 | 706.86 | 169.95 | 0.33 | 27.77 | 3.33 |
| A20-R0 | 259.37 | 257.71 | 0.00 | 848.00 | 140.8 | 565.49 | 169.95 | 0.33 | 0.00 | 3.33 |
| A20-R25 | 259.37 | 257.71 | 199.94 | 636.00 | 140.8 | 565.49 | 169.95 | 0.33 | 13.89 | 3.33 |
| A20-R50 | 259.37 | 257.71 | 399.89 | 424.00 | 140.8 | 565.49 | 169.95 | 0.33 | 27.77 | 3.33 |
| A40-R0 | 259.37 | 257.71 | 0.00 | 848 | 281.7 | 424.12 | 169.95 | 0.33 | 0.00 | 3.33 |
| A40-R25 | 259.37 | 257.71 | 199.94 | 636.00 | 281.7 | 424.12 | 169.95 | 0.33 | 13.89 | 3.33 |
| A40-R50 | 259.37 | 257.71 | 399.89 | 424.00 | 281.7 | 424.12 | 169.95 | 0.33 | 27.77 | 3.33 |
| A60-R0 | 259.37 | 257.71 | 0.00 | 848.00 | 422.6 | 282.74 | 169.95 | 0.33 | 0.00 | 3.33 |
| A60-R25 | 259.37 | 257.71 | 199.94 | 636.00 | 422.6 | 282.74 | 169.95 | 0.33 | 13.89 | 3.33 |
| A60-R50 | 259.37 | 257.71 | 399.89 | 424.00 | 422.6 | 282.74 | 169.95 | 0.33 | 27.77 | 3.33 |

Note: A: aeolian sand; R: recycled coarse aggregate; the number is the replacement ratio; WR: water reduction.

### 2.3. Test and Analysis Methods

The flow chart of the experiment in this test is shown in Figure 3.

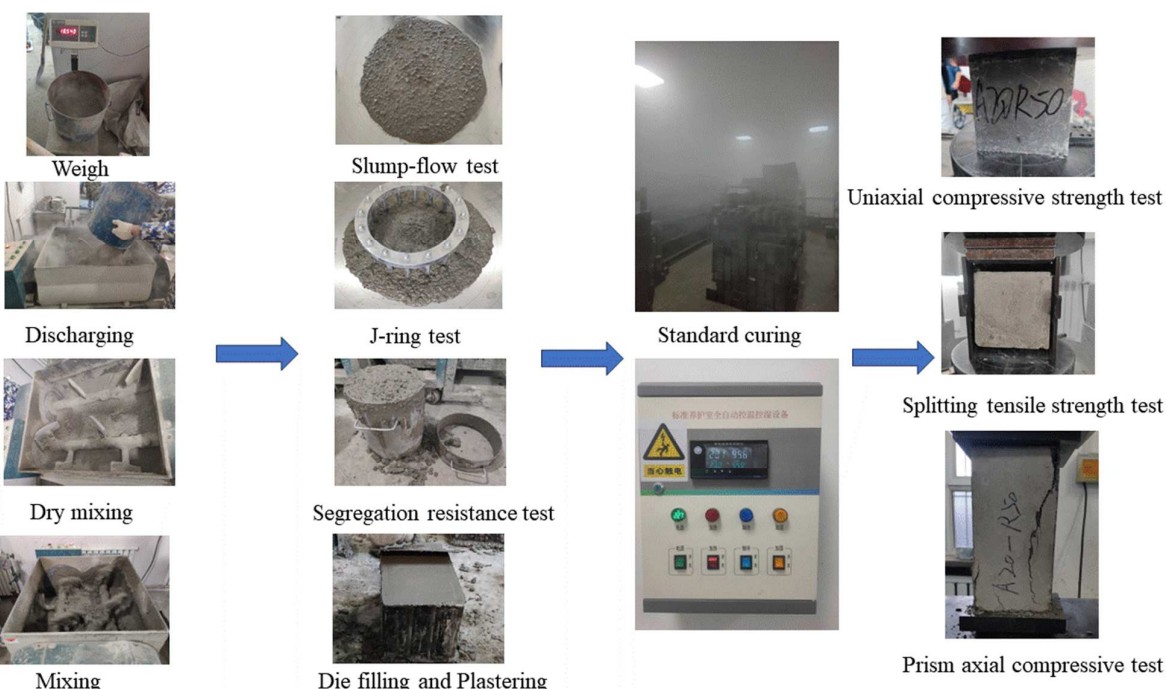

**Figure 3.** Flow chart.

### 2.3.1. Workability Tests

The workability test of fresh ARSCC was conducted in accordance with the standard (T/CECS 203-2021) [41]. The workability test displayed three typical behaviors: filling ability, passing ability, and segregation. The slump flow test was conducted to assess the filling ability of fresh ARSCC, while the J-ring test was employed to evaluate its passing ability. Additionally, segregation resistance was measured to determine the segregation of fresh ARSCC.

### 2.3.2. Mechanical Property Tests

Mechanical property tests were conducted on all ARSCC mixes in accordance with the standard (GB/T50081-2019) [42]. In this paper, the specimen sizes and ages used for each test are as follows: uniaxial compressive strength ($f_{cu}$) tests at the curing ages of 3, 7, 14, and 28 days to evaluate the compressive properties of concrete materials; splitting tensile strength ($f_t$) tests at the curing age of 28 days; and cube specimens (150 mm × 150 mm × 150 mm) were prepared for these two tests. Prism axial compressive ($f_c$) tests used prism specimens (150 mm × 150 mm × 300 mm) at the curing age of 28 days to evaluate the actual load-carrying capacity of a member under compression.

### 2.3.3. Scanning Electron Microscopy Test

Scanning electron microscopy (SEM) was used to examine the microstructure of the pore with a field emission scanning electron microscope (Hitachi S4800). Samples for analysis were collected from the heart of the cube. The fragments were sprayed with gold to improve their conductivity before testing.

### 2.3.4. Mercury Intrusion Porosimetry Test

Mercury intrusion porosimetry (MIP) was conducted on the part of ARSCC mixes in accordance with the standard (GB/T 21650.1-2008) [43] at the curing age of 28 days. MIP testing was conducted using an AutoPore Iv 9510 at a pressure range of 0.2 MPa to 415 MPa and used to measure the pore structure parameters of ARSCC. The test samples were approximately 1 cm$^3$ taken from the center of the cubic specimens and then soaked in anhydrous ethanol to prevent further hydration.

2.3.5. Range Analysis Method

The range analysis method is commonly used in orthogonal design [22]. The optimal results of an experiment can be judged by its method: the primary and secondary factors through simple calculation when experimental error is small. In obtaining the optimal blend for RCASCC, workability and mechanical properties were analyzed using the range method. The formula is as follows:

$$K_i = \sum_{i=1}^{n} x_i \tag{1}$$

where $K_i$ is the sum of $i$ elements under the factor P; $n$ is the number of tests conducted for corresponding influence factors; and $x_i$ is the indicator value of the test results at level $i$ under the corresponding influence factors ($i = 1, 2, 3, 4$):

$$\overline{K}_i = \frac{\sum_{i=1}^{n} K_i}{n} \tag{2}$$

$$R = \overline{K}_{i.max} - \overline{K}_{i,min} \tag{3}$$

where $\overline{K}_i$ is the combination of test indicators under a certain influence factor, and $R$ represents variation range.

The corresponding values of K in this paper are shown in Table 8.

**Table 8.** Table of factors.

| Factor | K$_1$ | K$_2$ | K$_3$ | K$_4$ |
|---|---|---|---|---|
| A | A0-R0<br>A0-R25<br>A0-R50 | A20-R0<br>A20-R25<br>A20-R50 | A40-R0<br>A40-R25<br>A40-R50 | A60-R0<br>A60-R25<br>A60-R50 |
| R | A0-R0<br>A20-R0<br>A40-R0<br>A60-R0 | A0-R25<br>A20-R25<br>A40-R25<br>A60-R25 | A0-R50<br>A20-R50<br>A40-R50<br>A60-R50 | -<br>-<br>-<br>- |

## 3. Results and Discussion

### 3.1. Workability Test Results

The experimental results of the study on the workability of ARSCC indicate that the incorporation of AS and RCA can adversely affect the workability of ARSCC, as shown in Figure 4. When AS and RCA are mixed into concrete at the same time, both have different degrees of water consumption, and a coupled water absorption effect occurs. By observing the state of the mix in Figure 4, it can be seen that under the condition of a constant water–cement ratio with a high replacement ratio of AS and RCA, the mix is drier and thicker, aggregates are slightly exposed, and the working performance is poorer.

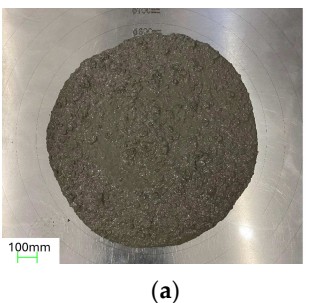 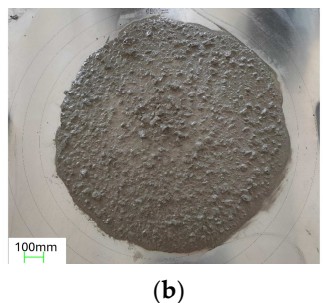 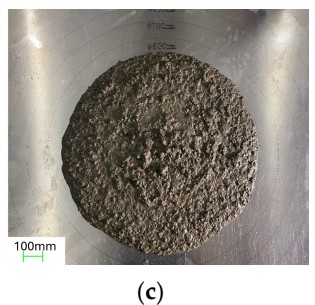

(a)        (b)        (c)

**Figure 4.** State of the mixture with (**a**) A0-R0, (**b**) A20-R50, and (**c**) A60-R50.

The slump-flow test results of ARSCC are shown in Figure 5. The slump-flow test examined the first of three key properties of ARSCC: filling ability. In this test, the range of slump-flow for all specimens was 510 mm–790 mm, and the control group A0-R0 had a slump-flow of 745 mm. The maximum slump-flow was 790 mm for A20-R0, higher by 6.04% than A0-R0. The effect of the RCA replacement ratio on slump-flow is shown in Figure 5a. Among the different AS replacement ratios, the slump-flow decreases with increasing RCA replacement ratio. The slump-flows were 8.72%, 10.76%, 12.67%, and 25.00% smaller with increasing RCA substitution at AS replacement ratios of 0, 20%, 40%, and 60%, respectively. The effect of the AS replacement ratio on slump-flow is shown in Figure 5b. The results show that the slump-flow increases and then decreases with the increase in AS replacement ratio. The slump-flows were 6.04%, 3.56%, and 3.68% larger and 8.72%, 21.28%, and 25.00% smaller with the increase in the AS replacement ratio for RCA replacement ratios of 0, 25%, and 50%, respectively. The results show that the substitution of a moderate amount of AS is favorable to enhance the slump-flow, while the substitution of RCA is always unfavorable to the slump-flow.

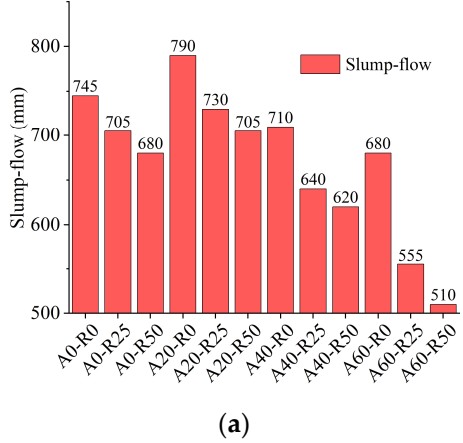
(a)

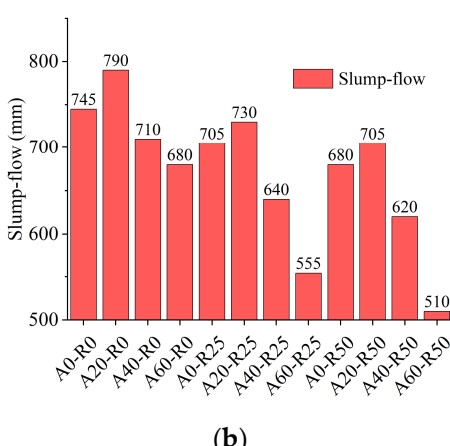
(b)

**Figure 5.** Slump-flows of ARSCC with a change in (**a**) RCA and (**b**) AS.

The J-ring test results of ARSCC are shown in Figure 6. The J-ring test examined the second of three key properties of ARSCC: passing ability. In this test, the range of the J-ring for all specimens was 0 mm–35 mm, and the control group A0-R0 had a J-ring of 20 mm. The maximum J-ring was 35 mm for A0-R50, higher by 6.04% than A0-R0. The effect of the RCA replacement ratio on the J-ring is shown in Figure 6a. Among the different AS replacement ratios, the J-ring increases with increasing RCA replacement ratio. The J-ring values were 42.86%, 66.67%, 50.00%, and 0% higher with increasing RCA substitution at AS replacement ratios of 0, 20%, 40%, and 60%, respectively. The effect of the AS replacement ratio on the J-ring is shown in Figure 6b. The results show that the J-ring decreases with the increase in the AS replacement ratio. The J-ring values were 100%, 80%, and 100% smaller with the increase in AS replacement ratio for RCA replacement ratios of 0, 25%, and 50%, respectively. The results show that the substitution of RCA enhances the J-ring, while the substitution of AS decreases the J-ring.

The segregation resistance test results of ARSCC are shown in Figure 7. Segregation resistance is the third characteristic of fresh ARSCC. The effect of RCA and AS replacement ratios on segregation resistance is shown in Figure 7a,b. The substitution of both AS and RCA resulted in a decrease in the segregation resistance. A60-R50 showed the lowest segregation resistance of 1.9% compared to A0-R0, which was 74.60% lower. The results show that both AS and RCA substitutions decrease the segregation resistance of ARSCC.

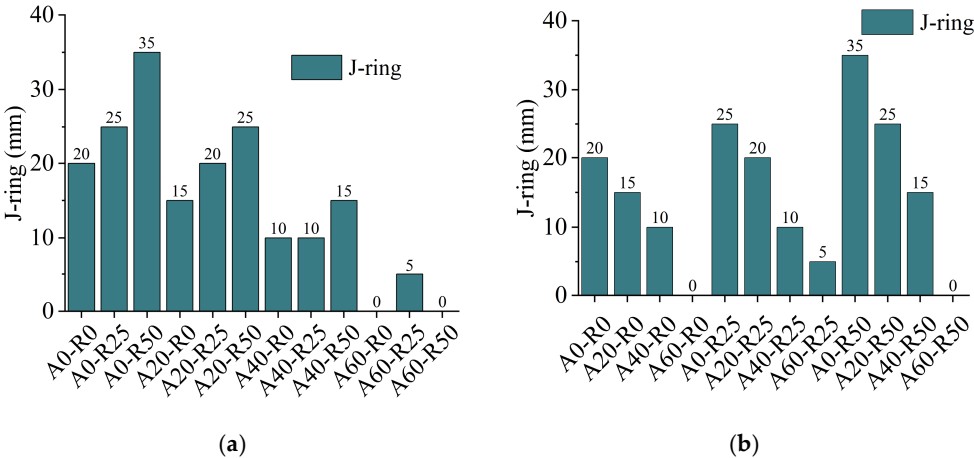

**Figure 6.** J-ring values of ARSCC with a change in (**a**) RCA and (**b**) AS.

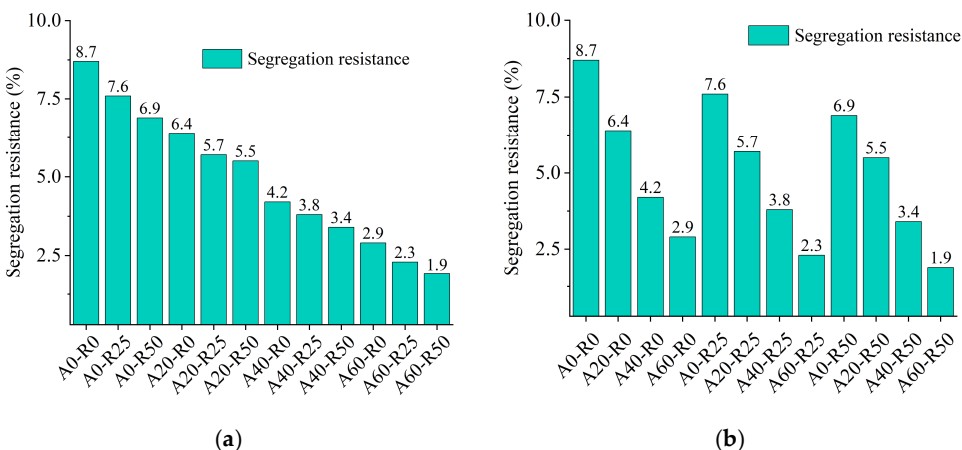

**Figure 7.** Segregation resistance of ARSCC with a change in (**a**) RCA and (**b**) AS.

These results indicate that the incorporation of AS and RCA in fresh concrete will have different effects on the different workability indices of ARSCC. According to Chu et al. [24], Leite et al. [19], and Safiuddin et al. [23], this behavior is due to two factors: The first observation is that RCA exhibits greater surface roughness after the crushing process compared to NCA, while AS has a larger specific surface area, which necessitates the use of more water. Second, the aeolian sand particles are finer than natural sand, the specific surface area is larger, and the water absorption is stronger. In addition, RCA is mostly irregular, similar to gravel aggregate. When the mix flows, its non-flat and smooth form is not conducive to the wrapping of cement slurry, which affects the mix segregation. The mixing of excessive recycled aggregates increases the rate of segregation and affects the compatibility of the mixture.

### 3.2. Analysis of Workability

The results of using the range method to calculate the effect of AS and RCA on the workability indexes are shown in Table 9. Comparing the R values under each evaluation index, A > R in slump-flow, J-ring, and segregation resistance. The results show that AS is the main factor in the workability of ARSCC. With an increase in the replacement ratio of aeolian sand and recycled aggregates, each factor shows a significant downward trend. However, considering the economic factors and environmental impact while ensuring the workability of the suitable mixture selection, A20-R25 and A20-R50 are chosen.

**Table 9.** Range method analysis of workability.

| Workability Test | Factors | $K_1$ | $K_2$ | $K_3$ | $K_4$ | $\overline{K}_1$ | $\overline{K}_2$ | $\overline{K}_3$ | $\overline{K}_4$ | R |
|---|---|---|---|---|---|---|---|---|---|---|
| Slump-flow | A | 1844 | 1800 | 1749 | 1623 | 614.67 | 600.00 | 583.00 | 541.00 | 73.67 |
| | R | 2440 | 2343 | 2233 | | 610.00 | 585.75 | 558.25 | | 51.75 |
| J-ring | A | 30 | 19 | 18 | 7 | 10.00 | 6.33 | 6.00 | 2.33 | 7.67 |
| | R | 33 | 25 | 16 | | 8.25 | 6.25 | 4.00 | | 4.25 |
| Segregation resistance | A | 42 | 26 | 21 | 12 | 26.93 | 27.13 | 26.27 | 24.70 | 10.00 |
| | R | 33 | 30 | 38 | | 8.25 | 7.50 | 9.50 | | 2.00 |

### 3.3. Mechanical Properties Test Result

Tests of uniaxial compressive strength after curing 3, 7, 14, and 28 days, axial compressive strength, and splitting tensile strength after curing 28 days yielded the results displayed in Table 10.

**Table 10.** Test result of mechanical properties.

| Mix Code | Mechanical Properties Test Results | | | | | |
|---|---|---|---|---|---|---|
| | 3 d $f_{cu}$ (MPa) | 7 d $f_{cu}$ (MPa) | 14 d $f_{cu}$ (MPa) | 28 d $f_{cu}$ (MPa) | 28 d $f_c$ (MPa) | 28 d $f_t$ (MPa) |
| A0-R0 | 14.2 | 20.7 | 27.2 | 33.0 | 24.8 | 4.18 |
| A0-R25 | 13.4 | 18.9 | 26.1 | 31.4 | 22.9 | 3.52 |
| A0-R50 | 14.5 | 23.9 | 27.5 | 34.2 | 26.0 | 3.36 |
| A20-R0 | 14.3 | 23.9 | 27.3 | 34.5 | 26.7 | 4.64 |
| A20-R25 | 14.2 | 24.5 | 26.0 | 33.5 | 25.1 | 3.76 |
| A20-R50 | 14.8 | 26.4 | 28.1 | 36.5 | 29.3 | 3.54 |
| A40-R0 | 14.2 | 20.8 | 27.0 | 32.5 | 24.1 | 3.46 |
| A40-R25 | 13.2 | 19.1 | 25.3 | 30.5 | 21.7 | 3.64 |
| A40-R50 | 14.6 | 19.8 | 26.5 | 32.2 | 22.1 | 3.30 |
| A60-R0 | 13.8 | 19.2 | 25.9 | 30.6 | 21.7 | 3.14 |
| A60-R25 | 13.1 | 17.5 | 23.4 | 29.6 | 20.4 | 3.02 |
| A60-R50 | 14.0 | 16.0 | 24.8 | 29.2 | 19.2 | 2.88 |

3.3.1. Uniaxial Compressive Strength

The effect of the replacement ratios of RCA and AS on the uniaxial compressive strength of ARSCC at different ages is shown in Figures 8 and 9. The uniaxial compressive strengths of ARSCC mixtures were evaluated at 3, 7, 14, and 28 days. At three days, an increase in the uniaxial compressive strength of ARSCC was observed. Compared with the reference, A20-R50 obtained the highest uniaxial compressive strength value at 4.22%, and A60-R25 exhibited a decrease at 7.75% compared with the control mixture (A0-R0). The results for days 7, 14, and 28 are similar to those of 3 days. For 7, 14, and 28 days, the maximum increase in uniaxial compressive strength of ARSCC is found in A20-R50 at 10.38%, 3.31%, and 5.52%, respectively, and the maximum decrease is exhibited by A60-R50 at 13.21% and 10.74% at 7 and 28 days, respectively, and A60-R25 at 13.9% at 14 days, compared with the control mixture (A0-R0).

From Figure 8, it can be concluded that the substitution of RCA decreases and then increases the uniaxial compressive strength in different curing days with different AS replacement ratios, but the different patterns exhibited in 7 d-A20, 7 d-A60, 28 d-A40, and 28 d-A60 may be related to the fabrication and curing of the specimens. R50 compared to R0 uniaxial compressive strength improved by 2.11%, 3.50%, 2.82%, and 1.45% at 3 days of maintenance, 15.46% at 7 days, 10.46%, 4.81%, 16.7%, 1.10%, 2.93%, 1.85%, and 4.24% at 14 days, and 3.64% at 28 days.

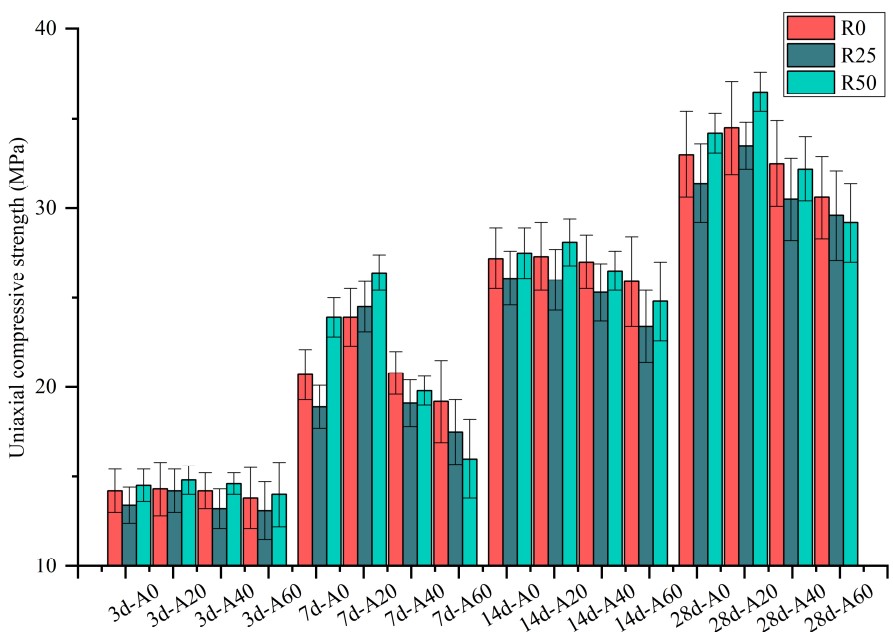

**Figure 8.** Uniaxial compressive strength of ARSCC with a change in RCA.

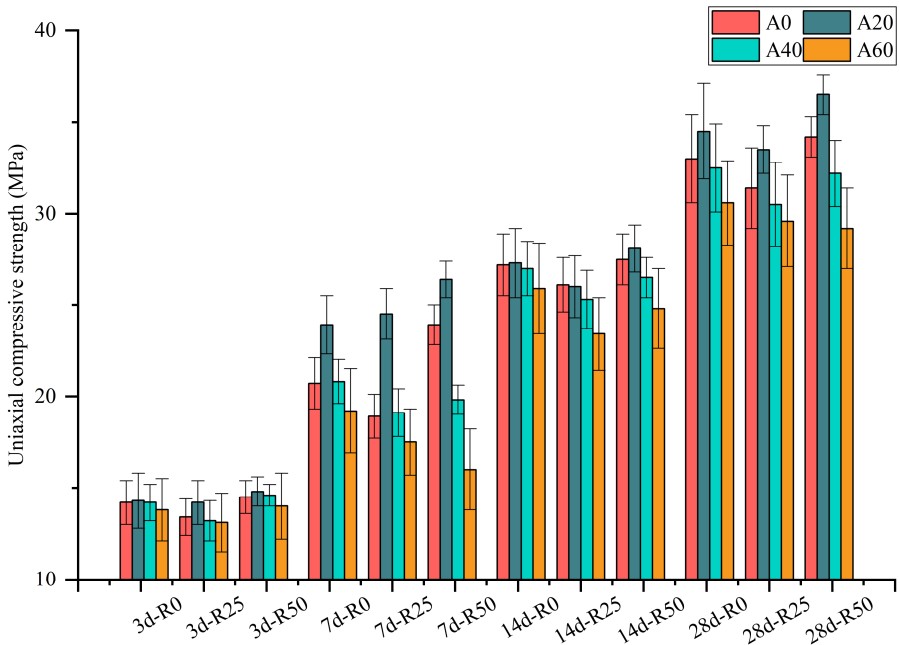

**Figure 9.** Uniaxial compressive strength of ARSCC with a change in AS.

From Figure 9, it can be concluded that the substitution of AS increases and then decreases the uniaxial compressive strength at different curing days with different RCA replacement ratios. The test results show that the uniaxial compressive strength decreases gradually with the increase in substitution rate in A20, A40, and A60, so the main discussion is on A20. Taking A0 compressive strength as a benchmark, the A20 uniaxial compressive strength change rate is 0.70%, 3.50%, and 5.59% at 3 days of curing, 15.46%, 29.63%, and 10.46% at 7 days, 0.37%, −0.38%, 1.85%, and 2.18% at 14 days, and 4.55%, 6.69%, and 6.73% at 28 days.

Among the various replacement ratios of AS and RCA in the concrete, the most significant strength development was observed with AS replacement at levels of 20–40% and RCA replacement at a level of 50%. Therefore, in terms of economic efficiency and

environmental impact, the combination of A20-R50 appears to be a suitable choice for replacement.

### 3.3.2. Splitting Tensile Strength

From Figure 10a,b, it can be seen that the effect of different AS and RCA replacement ratios on the splitting tensile strength of ARSCC is similar to their effect on compressive strength. When the replacement ratio of recycled aggregate is certain, splitting tensile strength tends to increase and then decrease with an increase in the replacement ratio, among which strength is higher at 20% of the replacement ratio of AS. The average increase of 0.02 MPa is about 0.7% compared with the strength of specimens at 0 replacement ratio. The pattern is that with an increase in recycled aggregate substitution, strength decreases and then increases. On the whole, when the replacement ratio of AS is 20% and the replacement ratio of RCA is 50%, the splitting tensile strength of the test pieces is the highest, which is 3.1 MPa, increased by 0.91 MPa. Regression analysis of uniaxial compressive strength and splitting tensile strength showed a clear power function relationship with the $R^2$ of 0.9122, as shown in Figure 11. The power function model is consistent with the results of Liu et al. [20].

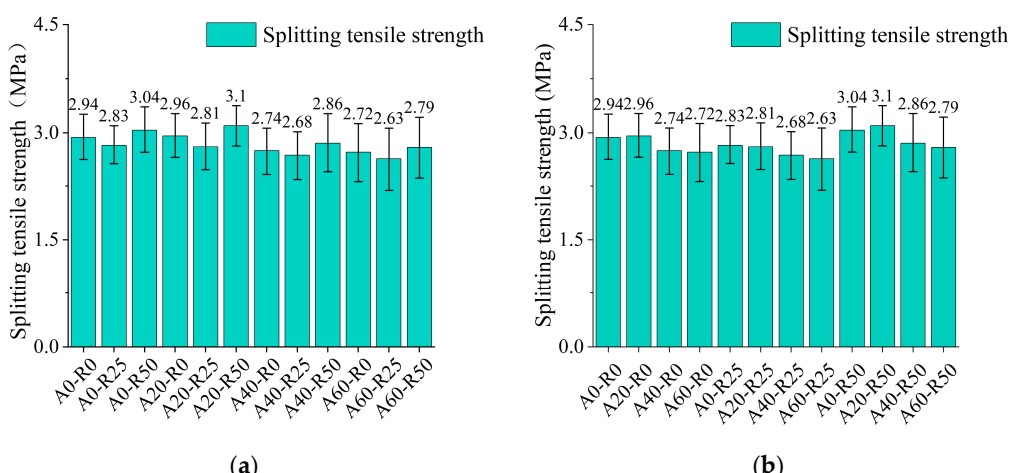

Figure 10. Splitting tensile strength with different replacement ratios of (a) RCA and (b) AS.

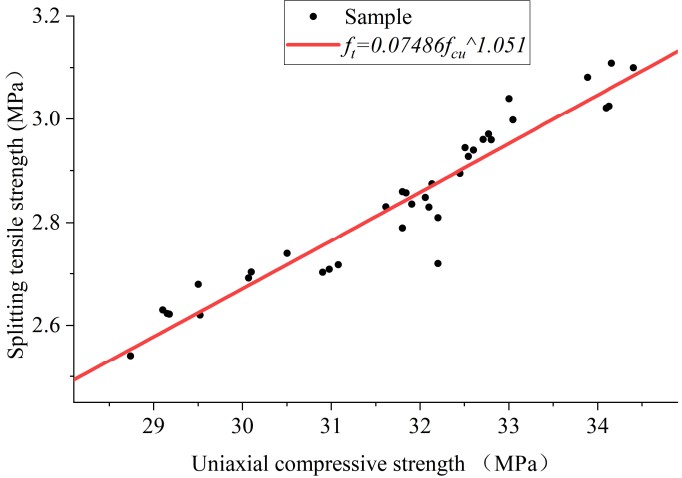

Figure 11. The correlation between splitting tensile strength and uniaxial compressive strength.

### 3.3.3. Static Stress–Strain Curve

By analyzing the load-displacement data collected from the tests, the stress–strain curves of the ARSCC were generated, and the results are presented in Figure 12. The

curves mainly feature ascending and descending phases, with a segment of the ascending phase displaying linear characteristics and the descending phase after the peak exhibiting a gentle decline. AS and RCA affect both phases. In Figure 12a,b, the effect of R50 on the stress–strain curve is favorable at a low replacement ratio of AS. In Figure 12c,d, the effect of recycled aggregate substitution on the stress–strain curve is negative at high AS substitution rates. The results show that ARSCC has good stiffness and plasticity in A20, in which the A20-R50 curve has a full shape, and the substitution of AS and RCA can lead to good deformation properties of SCC.

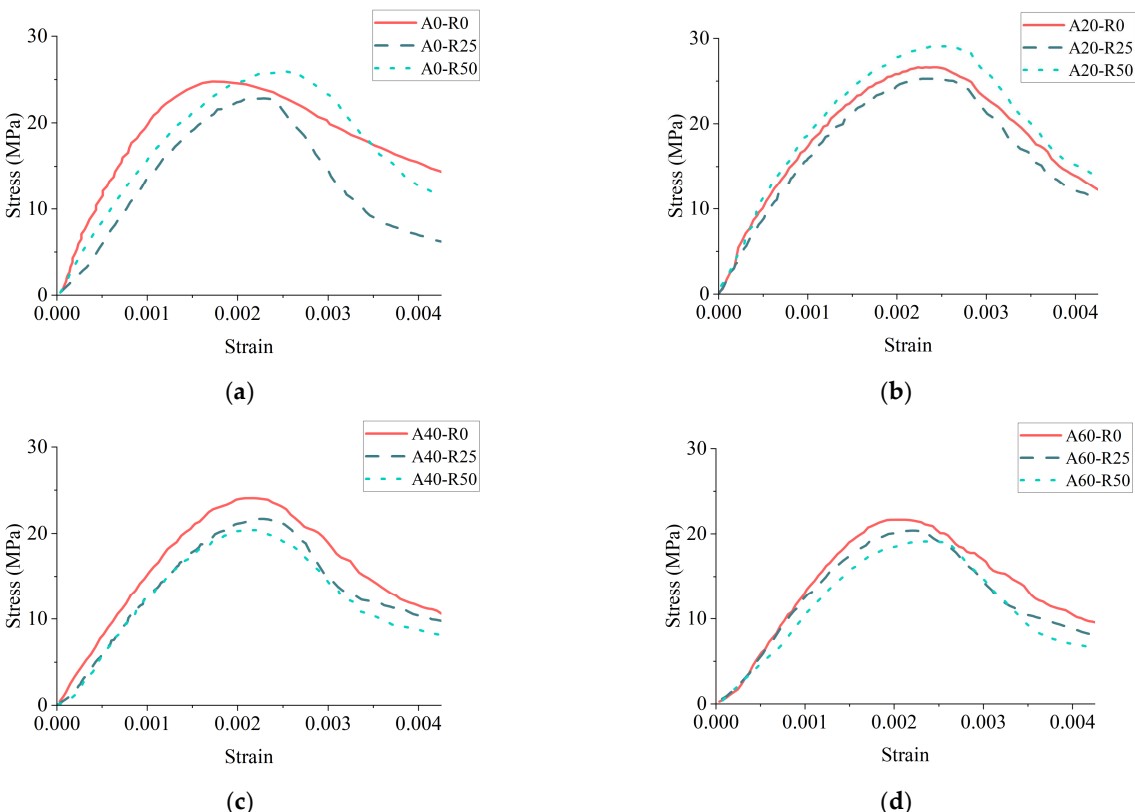

**Figure 12.** Complete stress–strain curves under axial compression for (**a**) A0, (**b**) A20, (**c**) A40, and (**d**) A60.

Guo [44] proposed the uniaxial compressive principal equation for ordinary concrete as early as the 1990s, and its expression is a segmental function with separate ascending and descending sections, and the *x*-axes and *y*-axes are dimensionless strain and stress, respectively, as shown in Equation (4).

$$\sigma_{nor} = \begin{cases} \alpha_1\varepsilon_{nor} + (3 - 2\alpha_1)\varepsilon_{nor}^2 + (\alpha_1 - 2)\varepsilon_{nor}^3, & 0 \leq x < 1 \\ \dfrac{x}{\left[\beta_1(\varepsilon_{nor}-1)^2 + \varepsilon_{nor}\right]}, & x \geq 1 \end{cases} \tag{4}$$

where $\varepsilon_{nor}$ is the normalized strain; $\sigma_{nor}$ is the normalized stress; and $\alpha_1$, $\beta_1$ are the parameters.

The dimensionless stress–strain curves of ARSCC in Table 11 show that the stress–strain curve shape of ARSCC is similar to ordinary concrete ($R^2 > 0.99$), so the uniaxial compressive principal equation of ordinary concrete (Equation (4)) can be considered to fit the stress–strain curve of ARSCC.

**Table 11.** Characteristic parameters of stress–strain curve.

| Mix Code | $\alpha_1$ | $\beta_1$ | $R^2$ |
|---|---|---|---|
| A0-R0 | 1.808 | 6.767 | 0.999 |
| A0-R25 | 1.930 | 3.776 | 0.998 |
| A0-R50 | 1.876 | 4.840 | 0.998 |
| A20-R0 | 1.593 | 5.686 | 0.998 |
| A20-R25 | 1.920 | 4.818 | 0.998 |
| A20-R50 | 1.706 | 3.908 | 0.992 |
| A40-R0 | 1.196 | 1.998 | 0.989 |
| A40-R25 | 1.561 | 7.732 | 0.998 |
| A40-R50 | 1.434 | 2.857 | 0.992 |
| A60-R0 | 1.257 | 1.906 | 0.994 |
| A60-R25 | 1.640 | 3.765 | 0.997 |
| A60-R50 | 1.821 | 2.925 | 0.996 |

The modulus of elasticity at the origin is chosen as the tangential young modulus ($E_t$) of ARSCC, and the results are shown in Figure 13. Figure 13a,b shows the effect of RCA and AS on the $E_t$, respectively. The $E_t$ of ARSCC decreases at high replacement ratios such as A40-R50 and A60-R50, decreasing by 68.21% and 51.83%, respectively, compared to A0-R0. While at low replacement ratios, such as A20-R25 and A20-R50, the modulus of elasticity of ARSCC increases by 16.14% and 38.55%, respectively.

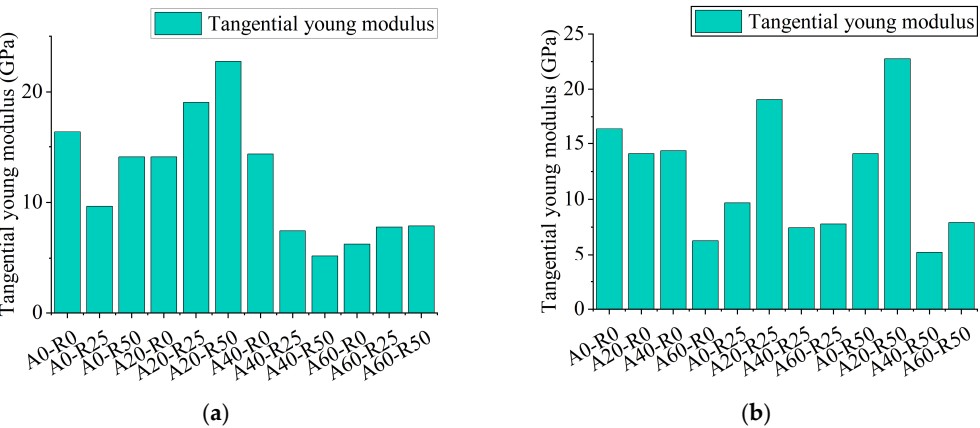

**Figure 13.** Tangential young modulus of ARSCC with a change in (**a**) RCA and (**b**) AS.

3.3.4. Axial Compressive Strength

The effect of the replacement ratios of AS and RCA on the axial compressive strength of ARSCC is shown in Figure 14a,b. The axial compressive strength of ARSCC ranged from 19.2 to 29.3 MPa. Compared to the value of 24.8 MPa with the control mixture (A0-R0), the maximum axial compressive strength increased by 18.14%, and the minimum one decreased by 22.58% with increased replacement ratios of AS and RCA.

The influence of the type of replacement ratio on the peak compressive strain of ARSCC is shown in Figure 15a,b. For a given replacement ratio of AS and a raised replacement ratio of RCA, higher peak compressive strain values were measured due to better hydration. For a given replacement ratio of RCA and a raised replacement ratio of AS, peak compressive strain values were increased and then decreased due to gradation being changed.

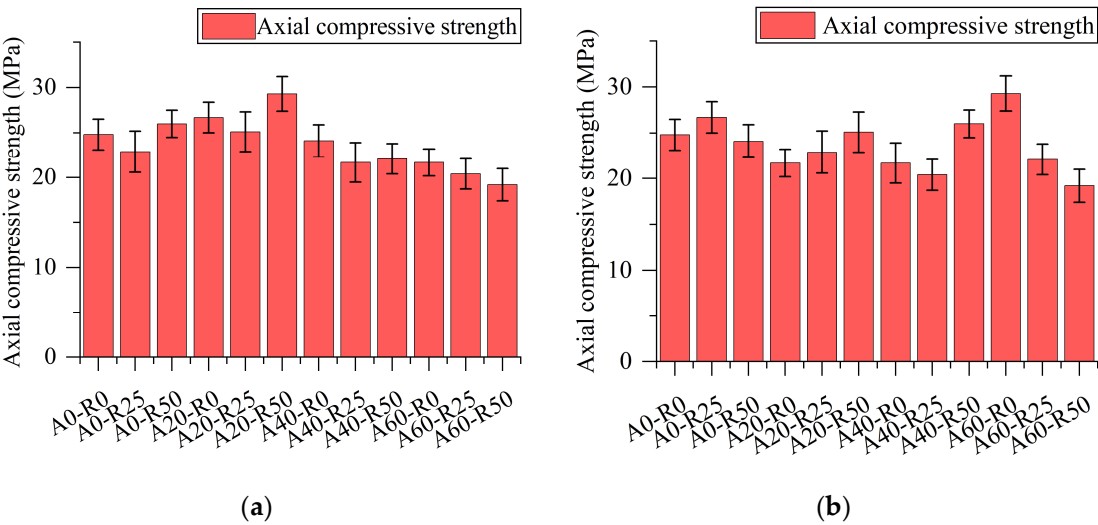

**Figure 14.** Axial compressive strength of ARSCC with a change in (**a**) RCA and (**b**) AS.

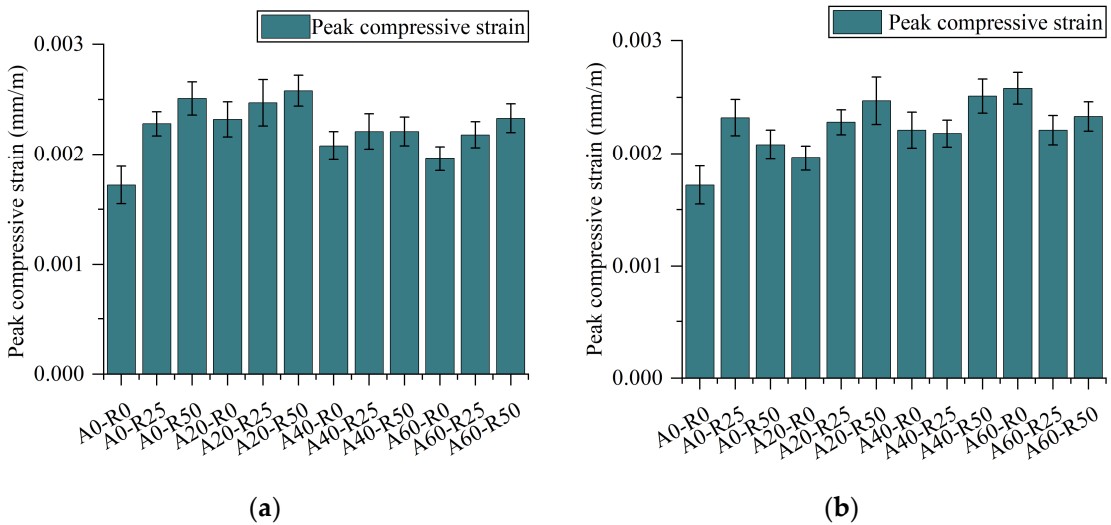

**Figure 15.** Peak compressive strain of ARSCC with a change in (**a**) RCA and (**b**) AS.

### 3.4. Analysis of Mechanical Properties

The results of using the range method to calculate the effect of AS and RCA on the mechanical property indexes are shown in Table 12. Comparison of the R values leads to the magnitude of the effect of AS and RCA on the relevant performance indexes. The order of influence of various factors on 7 d $f_{cu}$, 14 d $f_{cu}$, 28 d $f_{cu}$, 28 d $f_c$, and 28 d $f_t$ is as follows: AS > RCA. However, for 3 d $f_{cu}$, the order is RCA > AS. In the process of primary selection of optimal factors, RCA is well determined. With an increase in the replacement ratio of AS, R increases then decreases, and with an increase in the replacement ratio of RCA, R decreases then increases. Considering the economic factors and environmental impact while ensuring the mechanical properties of the suitable mixture selection, A20-R50 is chosen.

**Table 12.** Range method analysis of mechanical properties.

| Mechanical Properties Test | Factors | $K_1$ | $K_2$ | $K_3$ | $K_4$ | $\overline{K}_1$ | $\overline{K}_2$ | $\overline{K}_3$ | $\overline{K}_4$ | R |
|---|---|---|---|---|---|---|---|---|---|---|
| 3 d $f_{cu}$ | AS | 42.1 | 43.3 | 42 | 40.9 | 14.03 | 14.43 | 14.00 | 13.63 | 0.80 |
| | RCA | 56.5 | 53.9 | 57.9 | | 14.13 | 13.48 | 14.48 | | 1.00 |
| 7 d $f_{cu}$ | AS | 63.5 | 74.8 | 59.7 | 52.7 | 21.17 | 24.93 | 19.90 | 17.57 | 7.37 |
| | RCA | 84.6 | 80 | 86.1 | | 21.15 | 20.00 | 21.53 | | 1.53 |
| 14 d $f_{cu}$ | AS | 80.8 | 81.4 | 78.8 | 74.1 | 26.93 | 27.13 | 26.27 | 24.70 | 2.43 |
| | RCA | 107.4 | 100.8 | 106.9 | | 26.85 | 25.20 | 26.73 | | 1.65 |
| 28 d $f_{cu}$ | AS | 98.6 | 104.5 | 95.2 | 89.4 | 32.87 | 34.83 | 31.73 | 29.80 | 5.03 |
| | RCA | 130.6 | 125 | 132.1 | | 32.65 | 31.25 | 33.03 | | 1.78 |
| 28 d $f_c$ | AS | 73.7 | 81.1 | 67.9 | 61.3 | 24.57 | 27.03 | 22.63 | 20.43 | 6.60 |
| | RCA | 97.3 | 90.1 | 96.6 | | 24.33 | 22.53 | 24.15 | | 1.80 |
| 28 d $f_t$ | AS | 11.06 | 11.94 | 10.4 | 9.04 | 3.69 | 3.98 | 3.47 | 3.01 | 0.97 |
| | RCA | 15.42 | 13.94 | 13.08 | | 3.86 | 3.49 | 3.27 | | 0.59 |

## 4. Micro Test Result Analysis

### 4.1. Scanning Electron Microscopy

In this section, the microscopic analysis methods of SEM were further used to observe the microstructure, hydration products, and pore structure of ARSCC specimens under different replacement ratios after curing for 28 days.

Figure 16 shows SEM micrographs of ARSCC at 28 days of curing. The conventional self-compacting concrete (Figure 16a) had a dense microstructure with micro-cracks in the cement paste matrix or interfacial transition zone (ITZ). Figure 16a shows the irregular fine cracks and semi-encapsulated fly ash (FA) spherical particles distributed in the sample. The material attached to the surface of the aggregate in the form of small irregular particles is known as C-S-H gel. The exposed surface is relatively smooth. The hydration products shown in Figure 16b are more abundant but distributed with more holes of different sizes, and C-H-S gels exist near mesh-like cavities. The newly generated gels from cement particles are independent and relatively weakly linked with old mortar of already existing old products whose overall structure is laxer and shows porosity [45]. Figure 16c shows fewer pores and cracks distributed in lamellar old C-S-H gels. The comparison of microscopic morphology at three substitution rates clearly shows that the fine structure of concrete at a 25% RCA replacement ratio has more pores and cracks. Evidently, a concrete fine structure with a 25% RCA replacement ratio has more pore distribution and less integrity. The fine structure of ARSCC at a 50% replacement ratio, despite the presence of some pores, has an ITZ in close contact with aggregate and with a better bonding performance, as reflected in macro-mechanics given relatively higher compressive strength.

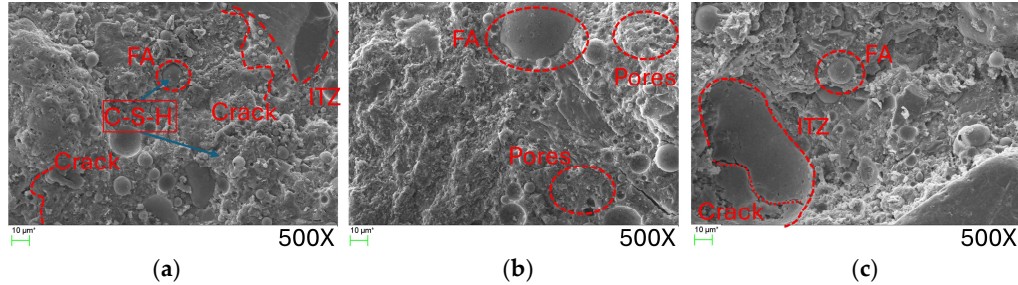

**Figure 16.** SEM micrographs of ARSCC with a 0 replacement ratio of AS. (**a**) A0-R0; (**b**) A0-R25; and (**c**) A0-R50.

In Figure 17, aggregate is almost invisible; a large number of irregular cement stones cover the surface of aggregate; small C-S-H gel particles can be observed; and almost no fly ash particles are found. Moreover, the degree of cement hydration is high, and fly ash particles may have been completely wrapped. However, semi-exposed spherical fly ash particles are shown in Figure 17b,c. In Figure 18, densely distributed holes and more cracks

are observed with the incorporation of AS. Further crack number and length development with increasing RCA replacement ratio is shown in Figure 18b,c. The surface irregularity of recycled coarse aggregate causes it to come more fully into contact with the cement mortar due to more adequate hydration of cement at the interface of recycled coarse aggregate concrete, and the gel is wrapped densely, exposing fly ash particles present around it [46]. The comparison between the denseness and integrity of interfaces of different specimens indicates that microstructure determines macroscopic performance at 0 and 50% of recycled aggregate replacement better than that at 25%, consistent with previous results of the cubic compressive strength test.

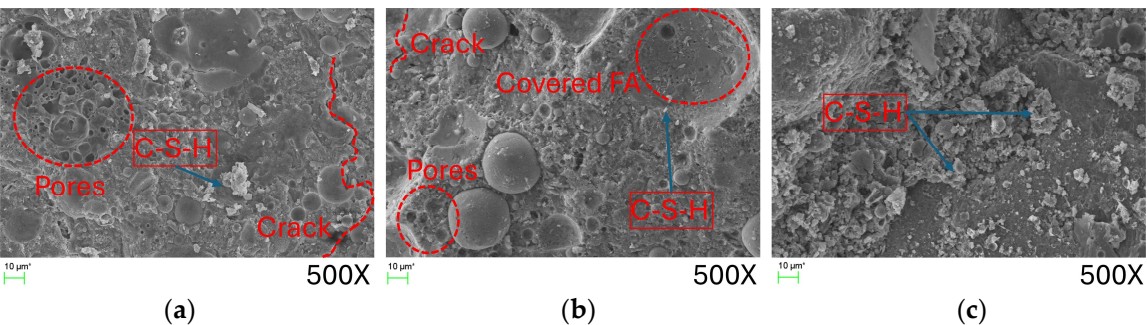

**Figure 17.** SEM micrographs of ARSCC with a 20% replacement ratio of AS. (**a**) A20-R0; (**b**) A20-R25; and (**c**) A20-R50.

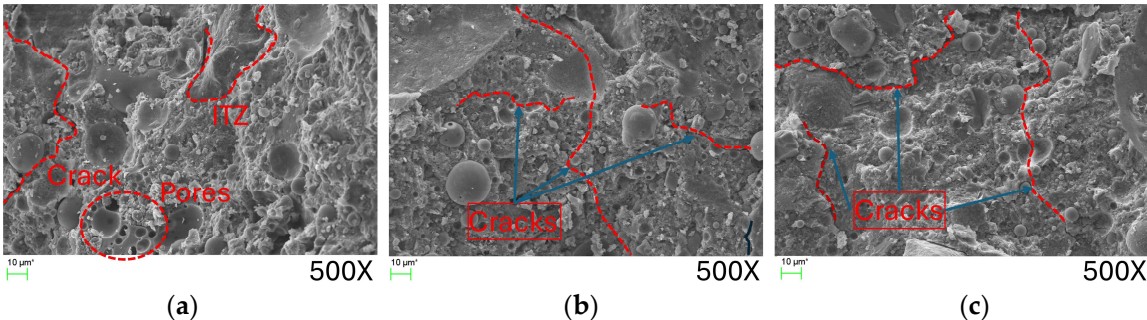

**Figure 18.** SEM micrographs of ARSCC with a 40% replacement ratio of AS. (**a**) A40-R0; (**b**) A40-R25; and (**c**) A40-R50.

*4.2. Mercury Intrusion Porosimetry*

In this section, the microscopic analysis method of MIP was used to observe the aperture distribution characteristics of four selected mixtures: A0-R25, A20-R25, A20-R50, and A40-R50 for tests.

Figure 19 presents pore size distributions of different replacement ratios of AS and RCA, and Table 13 summarizes corresponding pore characteristic parameters. According to the existing classification methods, the pore structure is divided into gel pores, transitional pores, capillary pores, and macro pores [47,48]. It is noticeable from Figure 19 that transitional pores were reduced with a suitable replacement ratio. Compared with A0-R0, with the increase in AS and RCA replacement ratios, the peaks first shift to the left and then to the right; that is, the distribution of the major pore size first increases and then decreases, but mainly in the category of transition pores. As for A40-R50, there are two obvious peaks, one of which belongs to the range of large pores, which may be due to the damage caused to the specimen during the sampling process, resulting in a certain macroscopic damage that has a certain impact on the test results.

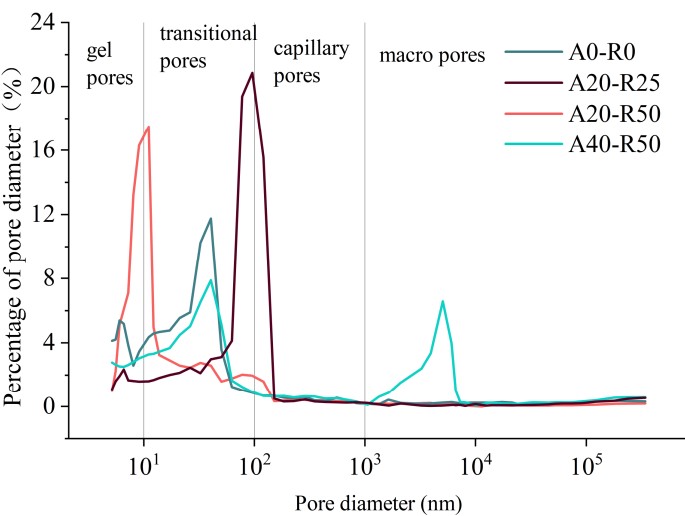

**Figure 19.** Pore size distribution of ARSCC with different replacement ratios.

**Table 13.** Pore characteristic parameters of ARSCC.

| Samples | Porosity (%) | Total Pore Area (nm²) | Average Pore Diameter (nm) | Intermediate Pore Diameter (nm) |
|---|---|---|---|---|
| A0-R0 | 20.8 | 43.3755 | 6.7107 | 21.11 |
| A20-R25 | 12.6 | 10.5872 | 1.5301 | 36.4 |
| A20-R50 | 10.3 | 8.1338 | 1.1192 | 50.4 |
| A40-R50 | 20.2 | 22.9284 | 2.3321 | 26.3 |

As illustrated in Table 13, porosity in the optimized replacement ratios of AS and RCA reduced by 50.48% from 20.82% to 10.3%. The average pore size was reduced by 83.38% from 6.7107 nm to 1.1192 nm. The results showed that the substitution of AS and RCA could reduce the porosity of SCC, enhance the compactness, and improve the internal pore structure. The improvement of pore structure by AS and RCA can be attributed to the following two aspects: (1) The substitution of AS improves the grade of the fine aggregate, fills a certain number of tiny holes, and reduces porosity. (2) Aeolian sand and recycled coarse aggregate reacted with calcium hydroxide in mortar to generate secondary C-S-H gels, which not only filled voids but also improved the strength of the mortar.

## 5. Conclusions

This study evaluated the fresh and hardened properties of SCC containing AS and RCA at different replacement ratios. By utilizing AS and RCA replacement ratios as variables, slump-flow, J-ring, and segregation resistance tests were performed to evaluate the workability of ARSCC, while cube compressive strength, split tensile strength, stress–strain curve, and axial compressive strength tests were conducted to evaluate its hardening properties. The microstructure and pore structure of ARSCC were analyzed using SEM and MIP tests. The following conclusions are drawn:

(1) The AS and RCA replacements significantly affected the workability of ARSCC. Range analysis revealed that AS had the greatest effect on workability.

(2) The AS and RCA replacements had positive and negative effects on the uniaxial compressive strength of concrete. The optimum replacement ratios of AS and RCA for the uniaxial compressive strength were 20% and 50%, respectively, resulting in a 4.22%, 10.38%, 3.31%, and 5.52% increase in the uniaxial compressive strength at 3, 7, 14, and 28 days, respectively.

(3) The effects of AS and RCA replacements on split tensile strength were also twofold, and the optimal replacement ratio of A20-R50 increased the split tensile strength by

5.44%. The split tensile strength and compressive strength were well correlated, with a correlation coefficient of 0.9122.

(4)  The regression equations of the stress–strain curves of ARSCC were in a good relationship, with all regression correlation coefficients greater than 0.99. AS and RCA had different effects on the modulus of elasticity, peak stress, and axial compressive strength. There were different degrees of improvement in A20-R50.

(5)  AS and RCA improved the microstructure and altered the pore structure of ARSCC. The changes in microstructure and pore structure corresponded to the variations in mechanical properties of ARSCC.

**Author Contributions:** Conceptualization, H.Y.; Methodology, S.L.; Software, F.H.; Formal analysis, S.Z. and F.H.; Resources, Q.L. and T.H.; Data curation, Q.L., T.H. and H.Y.; Writing—original draft, S.Z.; Writing—review & editing, S.Z.; Supervision, S.L.; Project administration, Q.L. All authors have read and agreed to the published version of the manuscript.

**Funding:** This research was funded by [Study on Deterioration Mechanism of Aeolian Sand Recycle Aggregate Self-Compacting Concrete in Cold and Saline Area, the National Natural Science Foundation of China] grant number [52168037] and [Key Research and Development Program Plan Project for Projects of Xinjiang Uygur Autonomous Region] grant number [2022B03036].

**Data Availability Statement:** The data presented in this study are available on request from the corresponding author.

**Conflicts of Interest:** The authors declare no conflict of interest.

## Acronym List

| | |
|---|---|
| A | Aeolian sand |
| R | Recycled coarse aggregate |
| AS | Aeolian sand |
| RCA | Recycled coarse aggregate |
| SCC | Self-compacting concrete |
| ARSCC | Aeolian sand recycled coarse aggregate self-compacting concrete |
| $f_{cu}$ | Uniaxial compressive strength |
| $f_t$ | Splitting tensile strength |
| $f_c$ | Prism axial compressive |
| $E_t$ | Tangential young modulus |

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
