# Peer review of "Basic Mechanical Properties of Self-Compacting Concrete Prepared with Aeolian Sand and Recycled Coarse Aggregate"

_buildings, doi:10.3390/buildings14092949_

Round 1
Reviewer 1 Report
Comments and Suggestions for Authors
1. The significant and necessity of investigation on the mechanical properties of self-compacting concrete containing aeolian sand and recycled coarse aggregate shall be further highlighted.
2. The influencing mechanism from the incorporation of aeolian sand and recycled coarse aggregate on the self-compacting concrete properties shall be clearly analyzed and elucidated, especially as mentioned in line 236-237, “When AS and RCA are mixed into concrete at the same time, both have different degrees of water consumption, and a coupled water absorption effect occurs.”
3. The effect from different pore structure of self-compact concrete containing different amounts of aeolian sand and recycled coarse aggregate may require further comprehensive investigation and analysis.
4. Section Conclusion may require some conciseness and refinement.
Comments on the Quality of English LanguageSome polish and improvement on the language expression may be required.
Author Response
Thank you for the proposed changes, the revised response is sent as an attachment. Please see the attachment.

Reviewer 2 Report
Comments and Suggestions for Authors
Dear authors
Thank you for submitting your manuscript entitled "Basic Mechanical Properties of Self-Compacting Concrete Prepared with Aeolian Sand and Recycled Coarse Aggregate".
Although the manuscript has lots of experimental test results, several parts require major revisions. With its present form, it seems that the manuscript is not suitable for publication. You can find some comments which can help you improve the quality of the manuscript.
General comments
1- The quality of English is not satisfactory for publication. In this context, a thorough proofreading is required.
2- Some typing and grammatical errors are detected in the manuscript
Examples:
-However, Large amounts of natural (line 25)
- it can become the main fine aggregate of concrete (line 54)
- Characteristic properties of Portland cement are 81 tabulated in Table 1, the chemical composition of fly ash in Table 2, and performance parameters in Table 3. (line81, 82)
- AS is extra fine sand their incorporation into concrete has a significant 97 effect on performance of SCC
- Recycled concrete aggregates (RCA) were obtained from concrete roads that was demolished in Urumqi and natural concrete aggregates (NCA) is taken from Urumqi West Construction Ltd
- It is generally accepted from literature that amount of attached mortar to coarse recycled aggregate is lower than fine recycled aggregate [21,35].
- And analysis samples were collected within heart of the cube. The fragments were sprayed with gold to improve 161 their conductivity before the test.
3- In Table 3, you have defined the term "Fineness". Is it finenes modulus? If so, you can specify it in the materials and methods section. The mathematical formulation of the fineness modulus should also be presented in the relevant section.
4- In Figure 1 and Figure 2, the definition of the y-axis (cumulative sieve residue percentage %) should be defined as cumulative undersize (%)
5- The interpretation of Figure 3 is confusing. You should clearly explain your results given in Workability Test Results.
6- The comments on Figure 4 - 6 should also be refined based on your fundamental observations/findings.
7- In Figures 7 and 8, you have defined a compressive strength. I guess you are talking about the uniaxial compressive strength (UCS). If so, you should correct it properly in the whole manuscript.
8- Lines 303 - 313 should be refined. They are not clearly explained.
9- When it comes to the stress-strain relationships indicated in Figure 11, you should also calculate the tangential young modulus (Et) of different concrete mixtures by considering these curves. The variations in the tangential young modulus should also be discussed in detail.
10- The explanations in Figure 11 should be defined in more detail in this table. For example, how did you normalize the stress and strain?
11- You should discuss the variations in the uniaxial compressive strength (UCS) of concrete mixtures by considering your experimental design. It is very important to highlight the importance of your manuscript.
12- The title of Table 6 should be defined as "physical and mechanical properties of the coarse aggregates". Herein, in Table 6, the terms "Crush index (%)" and "robustness (%)" should be defined with their mathematical formulations in the materials and methods section.
13- In section 2.2, how did you establish the mixture design? Did you use the design of the experiment? or your mixture design is based on your previous experiences? They should be explained in detail.
14- There are lots of abbreviations in the manuscript. You should prepare a list of abbreviations at the end of the manuscript. All abbreviations should be defined in this list.
15- In the materials and methods section, you should give an appropriate flow chart showing your experimental procedure. You have lots of experimental procedures and it is hard to find them with their standards.
16- You need appropriate scale bars in some figures (Fig 3, Figure 14, Figure 15, Figure 16).
17- In Table 12, the terms K1-K4 should be explained in more detail. They are confusing and not understandable.
18- The abstract and conclusion sections should be rehandled both proofreading and considering the comments raised above.
19- The similarity index (iThenticate report) should be reduced at least to 15%.
I hope the comments would be beneficial in order to increase the quality of the manuscript.
With kind regards.
Comments on the Quality of English LanguageThe quality of English is not satisfactory for publication. The manuscript should be reconsidered by thorough proofreading.
Author Response

(The authors gave the same response as above.)

Round 2
Reviewer 1 Report
Comments and Suggestions for Authors
This paper can be accepted at current state.
Comments on the Quality of English LanguageSome further language improvement may be required.
Author Response
Comment 1 :Some further language improvement may be required.
Response 1: Thank you for pointing it out. We have done language editing.
Reviewer 2 Report
Comments and Suggestions for Authors
Dear authors
Thank you very much for revising the paper entitled "Basic Mechanical Properties of Self-Compacting Concrete Prepared with Aeolian Sand and Recycled Coarse Aggregate".
With its present form, the paper can be accepted for publication in Buildings. However, the authors should indicate the difference between uniaxial compressive strength and axial compressive strength. What is the difference between them? I guess, there is a misunderstanding.
Please make it clear before publication.
Kind regards
Author Response
Comment 1: With its present form, the paper can be accepted for publication in Buildings. However, the authors should indicate the difference between uniaxial compressive strength and axial compressive strength. What is the difference between them? I guess, there is a misunderstanding.
Response 1 : Thank you for pointing it out. We have added in manuscript as shown in lines 143, 146 and 147.